# Characterization of neurobehavioral pattern in a zebrafish 1-methyl-4-phenyl-1,2,3,6-tetrahydropyridine (MPTP)-induced model: A 96-hour behavioral study

**Khairiah Razali**[1], **Mohd Hamzah Mohd Nasir**[2], **Noratikah Othman**[3], **Abd Almonem Doolaanea**[4], **Jaya Kumar**[5], **Wisam Nabeel Ibrahim**[6,7], **Wael M. Y. Mohamed**[1,8]*

1 Department of Basic Medical Sciences, Kulliyyah of Medicine, International Islamic University Malaysia, Kuantan, Pahang, Malaysia, 2 Department of Biotechnology, Kulliyyah of Sciences, International Islamic University Malaysia, Kuantan, Pahang, Malaysia, 3 Department of Basic Medical Sciences, Kulliyyah of Nursing, International Islamic University Malaysia, Kuantan, Pahang, Malaysia, 4 Department of Pharmaceutical Technology, Kulliyyah of Pharmacy, International Islamic University Malaysia, Kuantan, Pahang, Malaysia, 5 Department of Physiology, Faculty of Medicine, UKM Medical Centre, Kuala Lumpur, Malaysia, 6 Department of Biomedical Sciences, College of Health Sciences, QU Health, Qatar University, Doha, Qatar, 7 Biomedical and Pharmaceutical Research Unit, QU Health, Qatar University, Doha, Qatar, 8 Clinical Pharmacology Department, Menoufia Medical School, Menoufia University, Shebeen El-Kom, Menoufia, Egypt

* wmy107@gmail.com

**Data Availability Statement:** All relevant data are within the paper.

## Abstract

Parkinson's disease (PD) is the most common brain motor disorder, characterized by a substantial loss of dopaminergic neurons in the substantia nigra pars compacta (SNpc). Motor impairments, such as dyskinesia, bradykinesia, and resting tremors, are the hallmarks of PD. Despite ongoing research, the exact PD pathogenesis remains elusive due to the disease intricacy and difficulty in conducting human studies. Zebrafish (*Danio rerio*) has emerged as an ideal model for researching PD pathophysiology. Even though 1-methyl-4-phenyl-1,2,3,6-tetrahydropyridine (MPTP) has been used to induce PD in zebrafish, behavioural findings are frequently limited to a single time point (24 hours post-injection). In this sense, we aim to demonstrate the effects of MPTP on zebrafish swimming behaviour at multiple time points. We administered a single dosage of MPTP (200μg/g bw) via intraperitoneal injection (i/p) and assessed the locomotor activity and swimming pattern at 0h, 24h, and 96h post-injection through an open field test. Analysis of the behaviour revealed significant reductions in swimming velocity (cm/s) and distance travelled (cm), concurrent with an increase in freezing maintenance (duration and bouts) in zebrafish injected with MPTP. In addition, the MPTP-injected zebrafish exhibited complex swimming patterns, as measured by the turn angle, meander, and angular velocity, and showed abnormal swimming phenotypes, including freezing, looping, and erratic movement. To conclude, MPTP administration into adult zebrafish induced hypolocomotion and elicited motor incoordination. Plus, the effects of MPTP were observable 24 hours after the injection and still detectable 96 hours later. These findings contribute to the understanding of MPTP effects on adult zebrafish,

**Funding:** This study was supported by the Malaysian Ministry of Higher Education (MoHE) through the Fundamental Research Grant Scheme (ref. no.: FRGS19-125-0734) granted to Asst. Prof. Dr. Wael Mohamed.

**Competing interests:** he authors have declared that no competing interests exist.

particularly in terms of swimming behaviours, and may pave the way for a better understanding of the establishment of PD animal models in the future.

## Introduction

Parkinson's disease (PD) is the second commonest neurodegenerative disease after Alzheimer's disease (AD), and it is expected to become more prevalent by 2050. According to the most recent published statistics, the incidence and prevalence of PD fluctuate around 4.5 to 21.0 and 18 to 328 cases, respectively, per 100,000 individuals annually (approximately 10 million people globally) [1]. Mechanistically speaking, PD is characterized by a significant constant loss of dopaminergic neurons from the substantia nigra pars compacta (SNpc) [2, 3]. The SNpc, which is found in the midbrain, is important for motor control because it supplies dopamine neurotransmitters to the striatum via a projection known as the nigrostriatal dopaminergic pathway [4]. The loss of dopaminergic neuronal cells in PD pathophysiology disrupts the supply of dopamine to the striatum, impairing motor capacity. As a result, motor disorders such as bradykinesia, dyskinesia, and resting tremors are considered gold standards in PD diagnostics [3]. Despite extensive research concerning PD, much uncertainty still exists surrounding the triggers that cause the specific loss of nigral dopaminergic neurons.

The zebrafish, scientifically known as *Danio rerio*, is a useful animal model for neurological disease research. Zebrafish have made significant contributions to the understanding of the nervous system. This species has been used in studies on PD, AD, Amyotrophic Lateral Sclerosis (ALS), and Progressive Supranuclear Palsy (PSP), among others [5–7]. Genotypically, 70% of the genes associated with human diseases are functionally comparable to those of zebrafish, with 80% of the genes being positioned on the same chromosome and in the same order, demonstrating that the two species share a conserved synteny [2, 8, 9]. In terms of PD, the mapping of the zebrafish dopaminergic neural system is well established, making it an ideal model for studying the mechanisms affecting the development of PD [10]. Moreover, the advances of translational research have enabled the establishment of transgenic and neurotoxin-induced zebrafish models capable of causing PD-like symptoms in zebrafish, both genetically and phenotypically [2, 11]. With this regard, behavioural abnormalities, neurotoxic effects, expressions of associated PD genes and proteins, immunomodulation, and many other topics have all been studied using zebrafish as a PD model.

One of the most widely utilized neurotoxins to generate PD in animal models is the 1-methyl-4-phenyl-1,2,3,6-tetrahydropyridine (MPTP) [12]. Mechanistically, the MPTP penetrates the blood-brain barrier (BBB) and specifically enters nigral dopaminergic neurons in the form of toxic 1-methyl-4-phenylpyridinium ion (MPP+) via dopamine transporters, specifically destroying the neurons [13]. The diminished nigral dopaminergic neurons in the MPTP-induced zebrafish model has been reported to cause movement abnormalities that mimic PD-like dyskinesia and bradykinesia [5, 14]. Besides, studies have also reported on weakened olfactory [15], tactile senses [16], and visual impairments [17] in PD animal models upon MPTP induction.

Evidently, the effect of MPTP can be seen one day after the injection [18, 19]. Even though MPTP has been used to induce PD in zebrafish, findings are frequently limited to a single time point (24 hours post-injection). More research is needed to assess the initial trend of the MPTP effects on zebrafish to confirm that the effects are still sustained a few days after the injection. Hence, the current study aims to demonstrate the MPTP effects on adult zebrafish

swimming behaviour at multiple time points (0h, 24h, and 96h after injection) by assessing the locomotor activity and swimming pattern. We delivered the MPTP to the peritoneal cavity of the zebrafish via intraperitoneal injection (i/p). We then conducted an open field test to statistically compare the swimming behaviours between control (saline-injected) and MPTP-injected zebrafish. Most of the PD research utilizing neurotoxin-induced models aim to develop therapeutic treatments that can potentially relieve PD symptoms. Hence, to avoid potential result misinterpretation due to confounding factors, it is important ensure that the PD model is already sustained and stabilized prior to any therapeutic interventions. Our study sheds light on the initial trend of MPTP effect adult zebrafish, particularly in terms of the swimming behaviour. Findings from this study could pave the way for a better understanding of PD animal models establishment in the future.

## Materials and methods

### Instrumentation

A 31G needle attached to a 0.5ml insulin syringe (U-100 Insulin 0.5ml 31G X 5/16″, Terumo®, Tae Chang Industrial Co., Ltd., Gyeongsangbuk-do, Korea) was used to administer saline and MPTP into zebrafish peritoneal cavity. Realme RMX1851 camera (Shenzen, China) set at 720p resolution and 30fps frame rate was used to record the swimming behaviour of adult zebrafish. Pre-processing of the recorded videos was done using VideoPad Professional version 7.32 (NCH Software Inc., CO, USA) video editor software. Analysis of the recorded videos was done by Noldus EthoVision XT version 11.5 (Noldus Information Technology, Wageningen, The Netherlands) tracking software. Statistical analysis was performed using GraphPad Prism 9.00 for Windows (GraphPad Software, CA, USA).

### Reagents and materials

The zebrafish housing and experimental tanks were filled with filtered facility water. Otohime B2 (Marubeni Nisshin Feed Co., Ltd, Tokyo, Japan) was given to zebrafish as daily diet. The powdered 1-methyl-4-phenyl-1,2,3,6-tetrahydropyridine (MPTP) was purchased from Sigma-Aldrich, MO, USA (Cat# M0896). Crystalline powdered sodium chloride (C0753, HmbG Chemicals, Progressive Scientific Sdn. Bhd., Selangor, Malaysia) was dissolved and used as MPTP solvent and saline injection. API® MelaFix (Mars Fishcare North America Inc., PA, USA) anti-bacterial treatment was used to facilitate wound healing post-injection.

### Zebrafish husbandry

A total of 40 male and female wild-type zebrafish (*Danio rerio*) aged three to four months old with an average initial body weight of 0.9 ± 0.1g were used in this study. The zebrafish were purchased from a local breeder in Kuantan, Pahang, Malaysia. All zebrafish were housed inside a 10L acrylic fish aquarium sized 58.5 cm (L) x 28.0cm (W) x 36.0cm (H) located at the IIUM Central Research and Animal Facility (CREAM) and given at least 10 days to acclimatize prior to the start of the experiment. Both aquariums (housing and experimental) were filled with filtered facility water and kept at a constant temperature of 26 ± 2˚C, had air bubbles for aeration, and biofilter stones to facilitate the nitrate degradation process. White fluorescence lamps were used to provide illumination with a 14h-light to 10h-darkness cycle, compliant with standard zebrafish care guidelines [20]. The zebrafish were fed three times a day (09:00, 12:00, and 16:00).

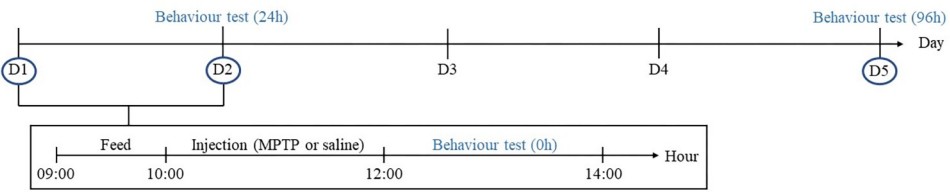

**Fig 1. Timeline of the experimental procedure.** MPTP injection was done on Day 1. Swimming behaviour tests were done on the 0th, 24th, and 96th hour after injection.

## Experimental design

Zebrafish were randomly divided into two groups (control and MPTP-injected groups), with each group consisting of 20 fish (n = 20). On the experimental days, zebrafish were fed one hour prior to the start of the procedure. Then, they were injected with either saline or MPTP depending on their assigned group. Swimming behaviour tests were performed at 0h, 24h, and 96h after injection. The initial and final weights (g) of each experimental zebrafish were recorded. Throughout the experimental period, zebrafish were housed individually, each in a 1.5L home tank sized 18.0cm (L) x 11.0cm (W) x 9.5cm (H). The entire experiment took five days to complete, and all zebrafish were sacrificed immediately after the completion of the study. The experimental paradigm of this study was permitted by the Institutional Animal Care and Use Committee (IACUC) under the International Islamic University Malaysia (IIUM), with reference to the animal ethics approval (ref. no.: IACUC-2021-02). The summarized timeline of the experimental design is illustrated in Fig 1.

## MPTP treatment

The powdered MPTP was dissolved with saline solution to prepare a 10mg/ml stock solution. MPTP is commonly administered into the adult zebrafish system via i/p injection or delivered directly into the brain through intracerebroventricular injection [21, 22]. In our study, we chose to administer the MPTP via i/p injection due to its proven rate of success and recovery as well as affordability [23, 24]. Prior to the injection, zebrafish were anesthetized with ice-cold water. One dose of 30μl of working solution containing 200μg MPTP per gram of body weight (200μg/g bw) was intraperitoneally administered into the zebrafish using a 31G needle attached to an insulin syringe (Fig 2). The point of the injection was at the abdominal cavity located posterior to the pelvic girdle, specifically at 45° to the base of the pelvic fin [23, 24]. The injection method was justified by looking for (1) mortality, (2) bleeding at the site of injection, (3) aberrant behaviour post-injection, (4) injection solution leakage, and (5) MPTP effect as critical indicators of injection success [24]. No documented mortality, bleeding, aberrant behaviour, and solution leakage, as well as the significant MPTP effect exhibited by the MPTP-injected group justified the success of the injection method. Control zebrafish were injected with 30μl of saline solution. Immediately after the injection, zebrafish were transferred into a recovery tank treated with anti-bacterial treatment to facilitate wound healing. All zebrafish were monitored upon recovery for any side effects or injuries.

## Swimming behaviour test

**Recording setup.** The recording setup was adapted from Selvaraj et al. [19] and modified where necessary. A 2.5L acrylic tank sized 24.0cm (L) x 13.5cm (W) x 13.5cm (H) was used as an assay tank. The tank was digitally divided into two imaginary zones, the inner IN zone and the outer OUT zone. The IN zone is a rectangular zone measuring 18cm (L) x 7.5 (W) located

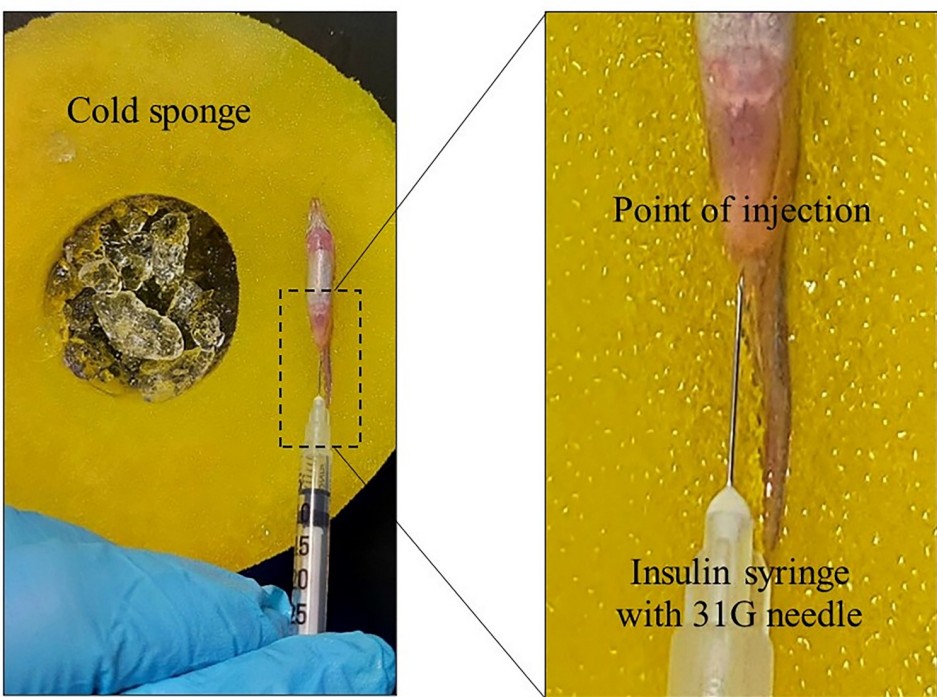

**Fig 2. Representative diagram of the experimental setup for intraperitoneal injection.** Anesthetized zebrafish was placed ventral side up on a cold sponge. The zoomed-in section depicts the point where the injection was given. A 31G needle attached to an insulin syringe was inserted into the injection point to deliver the MPTP (for MPTP-injected group) or saline solution (for control group).

at the centre of the assay tank while the OUT zone is located between the IN zone and the wall of the assay tank. The tank was placed on a stable surface and filled with filtered facility water until 8cm-height. The entire setup was walled with cardboard boxes and the experiment was conducted in an isolated room to minimize external visual and auditory noises. Illumination was supplied by LED lights. A camera was mounted on top of the tank to capture the top view. The recorded videos were saved in.mp4 format.

**Swimming behaviour recording.** A swimming behaviour test was performed every day from 0h until 96h post-injection. To minimize variances, the recording was executed at the same time (from 12.00 until 14:00) and in the same room each day. The test was performed on each zebrafish individually. Prior to the test, the zebrafish were acclimatized inside the assay tank for 5–10 minutes to familiarize themselves with the environment. After that, the zebrafish were let to swim in the tank for three minutes and their swimming behaviour was recorded throughout the swimming period. During the recording session, the experimenter monitored the zebrafish only through a monitor screen to prevent experimenter visibility and minimize experimenter effects. After the test, the zebrafish were immediately transferred back to their home tank. The recorded behaviours were analysed using Noldus EthoVision XT version 11.5 tracking software. Each parameter was compared between the control (saline-injected) and MPTP-injected groups.

**Behavioural analysis.** The swimming behaviour was defined by two measures: locomotor activity and swimming pattern [25]. The locomotor activity was evaluated in terms of swimming velocity (cm/s), total distance travelled (cm), and freezing maintenance (duration (s) and bouts), while the swimming pattern was examined in terms of the frequency and time spent (s)

in the IN zone as well as the turning rate (turn angle (˚), meander (˚/cm), and angular velocity (˚/s)).

The freezing bouts was defined as the number of total immobility (except for gills and eyes) in between swimming. The frequency and time spent in the IN zone were calculated by comparing the number of times visited and the duration swam in the inner zone to the outer zone. The turn angle was defined as the angle at which the head rotates during a turn. Meandering was defined as a movement that has no fixed direction or course. Angular velocity was defined as the amplitude and direction of zebrafish angular motion. Besides, the swimming phenotypes, which are the distinct types of swimming displayed by the zebrafish, were also investigated.

The recorded videos were pre-processed prior to analysis to reduce noise and errors. Then, the edited videos were imported to the Noldus EthoVision XT tracking software to analyse the swimming behaviour. The swimming velocity thresholds were set wherein the stop and start velocities for adult zebrafish are 0.20 and 0.21 cm/s, respectively. Velocity recorded below 0.20 cm/s were considered as non-swimming movements (for example slides, falls, sweeps) and freezing. Track smoothing (Lowess) profile was applied to the acquired tracking to minimize noise detection [Setup > Acquisition > Track Smoothing Profiles No filter > Smoothing (Lowess) > select] [26]. The tracking images captured were saved as.png files. The analysed data were exported as Excel files (.xlsx) and used for the subsequent statistical analysis.

## Statistical analysis

All parameters were analysed statistically through RM two-way ANOVA comparing the control and MPTP-injected groups, with Tukey's multiple comparisons test, as recommended by the statistical software. To ensure that the samples met the assumptions of the statistical approach, a normality test was performed prior to the RM two-way ANOVA. According to the Shapiro Wilk test for normality, both groups were normally distributed for all parameters. Aside from that, the samples met all other assumptions, including having continuous dependent variables, having an adequate sample size (as determined by G*Power software), and having similar relationship between the pairs of experimental conditions (as proven by Mauchly's test). The analysed data were presented as mean. P-values of less than 0.05 ($p < 0.05$) were considered significant for all statistical analyses.

## Results

### Effects of MPTP on adult zebrafish swimming behaviour

The alteration in the swimming behaviour of zebrafish after the i/p injection of MPTP neurotoxin was analysed. Several dosages of MPTP were evaluated in pilot experiments to determine the lowest observable effect level of the neurotoxin. 200µg of MPTP per gram body weight of zebrafish (200µg/g bw) elicited practically maximal decreases of general locomotor activity (swimming velocity and distance travelled) (data not shown), and thus, all subsequent investigations were conducted with this dose.

The zebrafish locomotor activity was markedly altered after the i/p injection of MPTP neurotoxin. Specifically, the alteration was already detected by 24h after MPTP injection and remained altered 96h after the injection. The swimming velocity of MPTP-injected zebrafish was decreased by 30% 24h after the injection and remained low on the 96th hour after the injection (Fig 3A). In terms of the distance travelled, the MPTP-injected zebrafish swam 30% shorter distances 24h and 96h after injection (Figs 3B and 4). Meanwhile, the control zebrafish maintained its swimming velocity and distance travelled throughout the three time-points, suggesting the effect of MPTP injection on zebrafish locomotor activity.

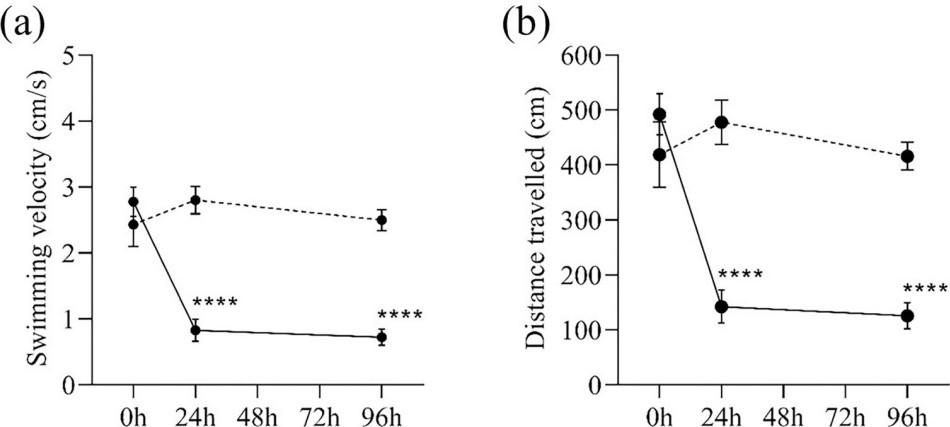

**Fig 3. General locomotor activity of MPTP-injected (solid lines) and control (dashed lines) zebrafish.** (a) Swimming velocity (in cm/s) and (b) Distance travelled (in cm) during the three-minute behaviour test at different time-points after the injection. Two-way RM ANOVA revealed that there were statistically significant interactions between the effects of MPTP injection and time on zebrafish swimming velocity ($F_{(2,76)} = 18.47$, $p < 0.0001$) and distance travelled ($F_{(2,76)} = 18.16$, $p < 0.0001$). Post-hoc Tukey's test found that the mean swimming velocity and distance travelled of MPTP-injected group were significantly different between 0h and 24h (velocity: $p < 0.0001$, 95% CI = [1.24, 2.67]; distance travelled: $p < 0.0001$, 95% CI = [241.10, 458.80]), and between 0h and 96h (velocity: $p < 0.0001$, 95% CI = [1.34, 2.77]; distance travelled: $p < 0.0001$, 95% CI = [265.80, 466.80]). All values are presented as mean and error bars represent the SEM. n = 20 per group. ****$p < 0.0001$.

During the three-minute behaviour test, the freezing maintenance (duration and bouts) of the MPTP-injected zebrafish was higher and more frequent than the control zebrafish. MPTP-injected zebrafish spent more than two minutes in immobile condition 24h and 96h after injection (Fig 5A). Notably, on the 24th hour after injection, MPTP-injected zebrafish alternated between swimming and freezing more frequently, as evidenced by the increased number of freezing bouts (Fig 5B).

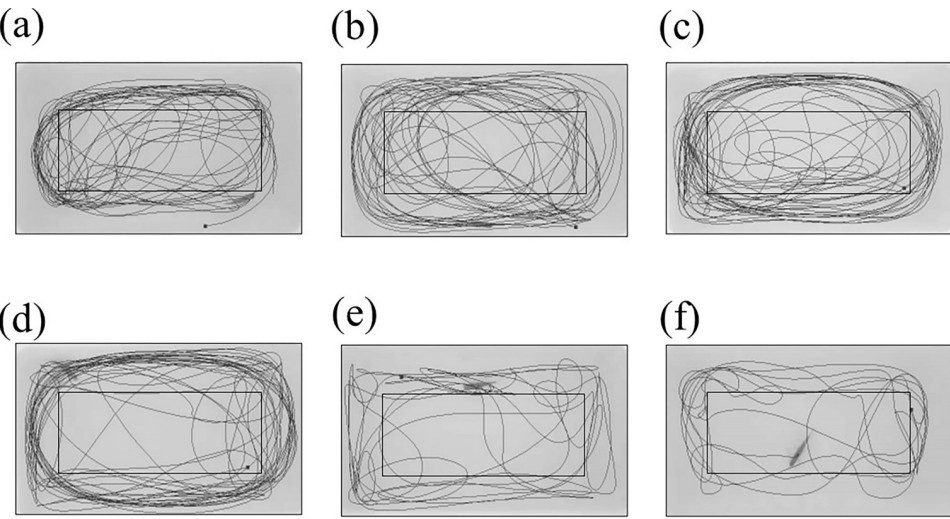

**Fig 4. Graphical representation of the swimming paths (in black) of the MPTP-injected and control zebrafish.** The outer rectangular border outlining the assay tank and the inner rectangle separate the IN and OUT zones. Each figure represents (a) Control, 0h. (b) Control, 24h. (c) Control, 96h. (d) MPTP-injected, 0h. (e) MPTP-injected, 24h. (f) MPTP-injected, 96h. The paths were generated by the EthoVision XT tracking software.

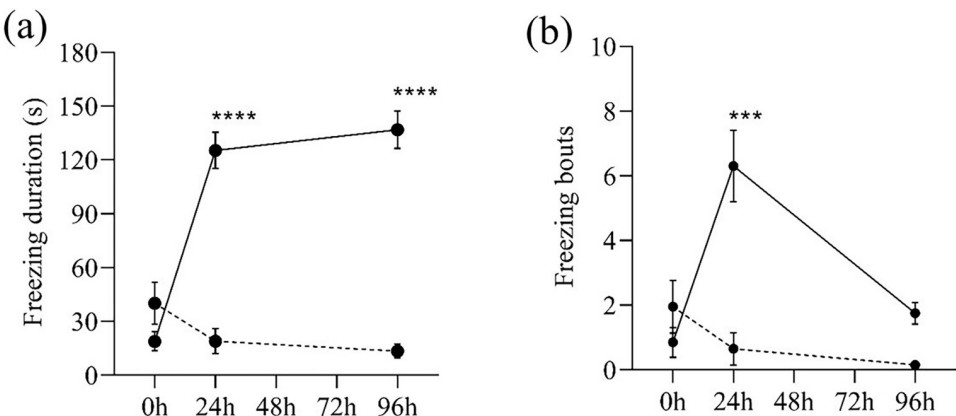

**Fig 5. Freezing maintenance of MPTP-injected (solid lines) and control (dashed lines) zebrafish.** (a) Freezing duration (in sec) and (b) Number of freezing bouts during the three-minute behaviour test. Two-way RM ANOVA revealed that there were statistically significant interactions between the effects of MPTP injection and time on the freezing maintenance (duration: [F (2,76) = 42.54, p <0.0001], bouts: [(F (2,76) = 15.46, p <0.0001)]. Post-hoc Tukey's test found that the mean freezing duration of MPTP-injected group were significantly different between 0h and 24h (p <0.0001, 95% CI = [-133.30, -79.69]), and between 0h and 96h (p <0.0001, 95% CI = [-147.00, -89.10]). Meanwhile, the mean number of freezing bouts of MPTP-injected group was significantly different between 0h and 24h (p = 0.0003, 95% CI = [-8.248, -2.652]). All values are presented as mean and error bars represent the SEM. n = 20 per group. ***p <0.001, ****p <0.0001.

The amount of time spent in the IN zone when compared to the OUT zone and the frequency with which it is visited is a type of swimming pattern that reflects the exploratory behaviour of a zebrafish. In our observation, although both the control and MPTP-injected zebrafish spent half of the time at the center of the assay tank and the other half circling the tank wall (Fig 6A), however, the MPTP-injected zebrafish appeared to switch between the center and the wall less frequently (Fig 6B), suggesting that they preferred to swim close to their initial position rather than exploring the surroundings.

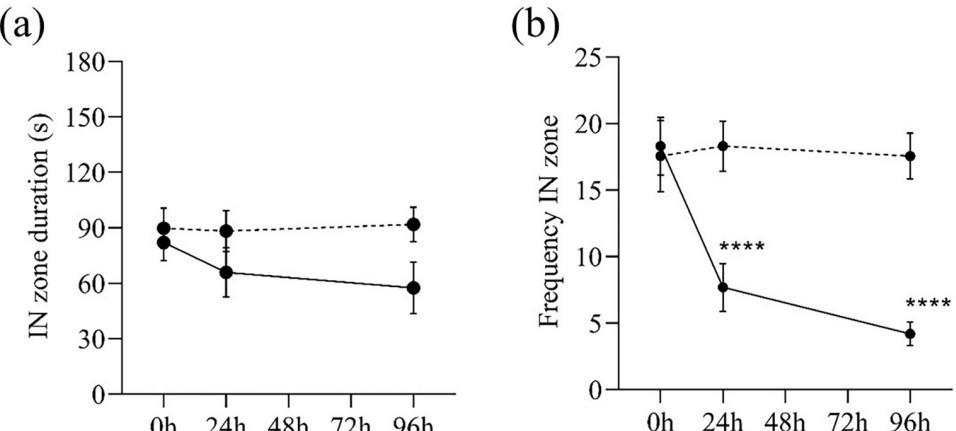

**Fig 6.** IN zone position of MPTP-injected (solid lines) and control (dashed lines) zebrafish (a) IN zone duration (in sec) and (b) Frequency in the IN zone during the three-minute behaviour test. Two-way RM ANOVA revealed that there was no statistically significant interaction between the effects of MPTP injection and time on the time spent in IN zone (F (2,76) = 0.85, p = 0.423). However, there was a significant interaction between the effects of MPTP injection and time on the frequency in IN zone (F (2,76) = 8.55, p = 0.0004). Post-hoc Tukey's test found that the mean frequency in IN zone of MPTP-injected group were significantly different between 0h and 24h (p <0.0001, 95% CI = [5.889, 15.31]), and between 0h and 96h (p<0.0001, 95% CI = [7.974, 20.23]). All values are presented as mean and error bars represent the SEM. n = 20 per group. ****p <0.0001.

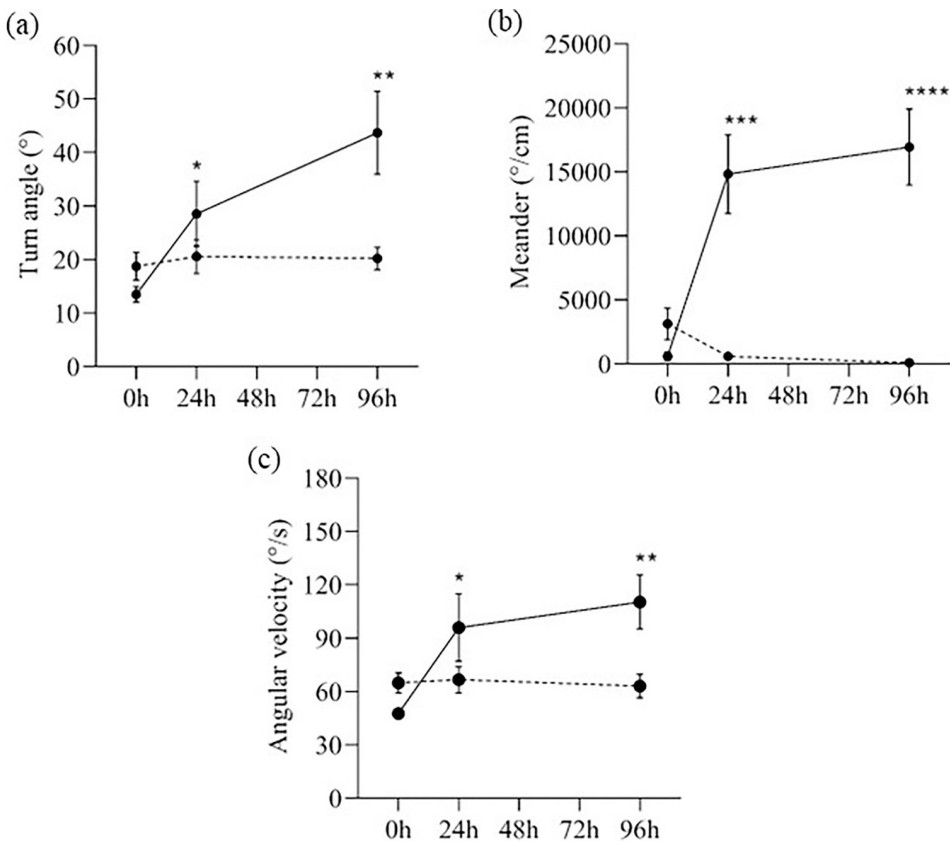

**Fig 7. Turning rate of MPTP-injected (solid lines) and control (dashed lines) zebrafish.** (a) Turn angle (in degree), (b) Meander (in degree/cm), and (c) Angular velocity (in degree/s) at different time-points after the injection. Two-way RM ANOVA revealed that there were statistically significant interactions between the effects of MPTP injection and time on meander (F (2,76) = 18.92, p < 0.0001), angular velocity (F (2,76) = 5.34, p = 0.007), and turn angle (F (2,76) = 6.40, p = 0.003). Post-hoc Tukey's test found that the mean meander, angular velocity, and turn angle of MPTP-injected group were significantly different between 0h and 24h (turn angle: p = 0.045, 95% CI = [-29.87, -0.2828]; meander: p = 0.0005, 95% CI = [-22021, -6436]; angular velocity: p = 0.047, 95% CI = [-96.09, -0.5698]), and between 0h and 96h (turn angle: p = 0.002, 95% CI = [-48.96, -11.42]; meander: p <0.0001, 95% CI = [-24002, -8704]; angular velocity: p = 0.001, 95% CI = [-99.85, -25.66]). All values are presented as mean and error bars represent the SEM. n = 20 per group. *p <0.05, **p <0.01, ***p <0.001, ****p<0.0001.

Another factor to describe the swimming pattern of the zebrafish is the turning rate, which is defined as the speed at which the fish can turn and is studied by measuring three parameters: meander, angular velocity, and turn angle. The MPTP-injected zebrafish had greater turn angles at 24h and 96h after injection (30˚- 40˚) than 0h and control zebrafish (~20˚) (Fig 7A). Corroborating this, the meander of MPTP-injected zebrafish, which is the change in direction of movement of a fish in relation to the distance moved by that fish, was three times greater at 24h and 96h after the injection than at 0h (Fig 7B), suggesting that the MPTP-injected fish turned at a greater degree per distance that it moved (˚/cm). Besides, the MPTP-injected zebrafish exhibited a higher angular velocity, as calculated by the change in direction of movement per unit time (˚/s). The rate of turn in direction of MPTP-injected zebrafish was nearly doubled 24h after injection and was increased by 50% on the 96th hour after injection when compared to 0h and control zebrafish (Fig 7C). Overall, MPTP-injected zebrafish had a higher turning rate 24h after injection and remained high 96 hours later.

Although a reduction in locomotor activity is the established hallmark for neurotoxin-induced zebrafish neuropathic behaviour, the swimming phenotypes should also be

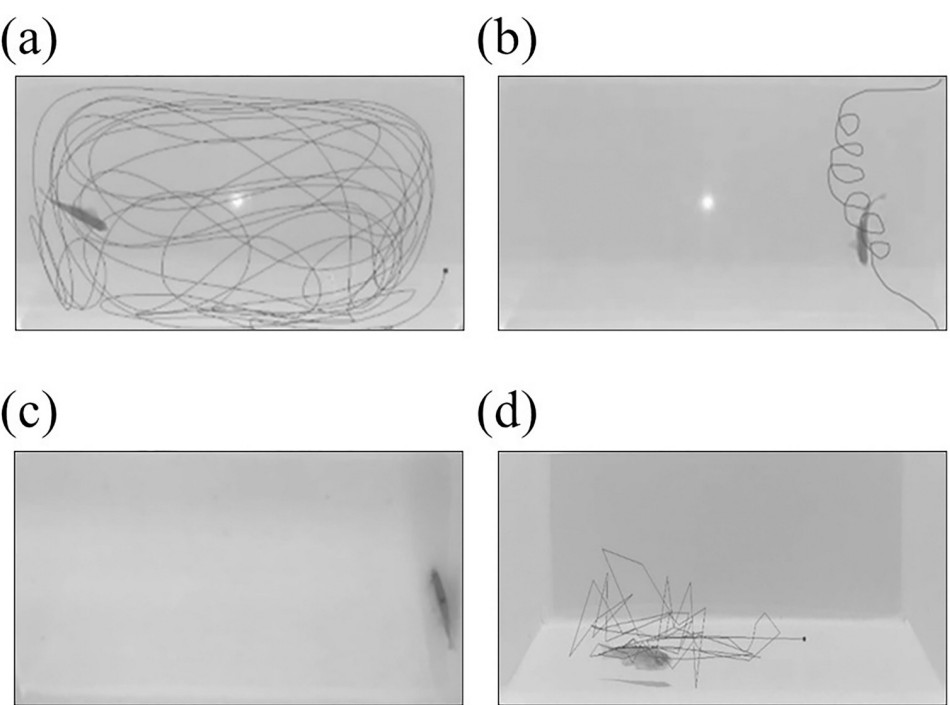

**Fig 8. Graphical representation of the swimming paths (in black) during the three-minute behaviour test.** Each figure represents distinct swimming phenotype, (A) normal continuous swimming (top view), (B) looping (top view), (C) freezing (top view), and (D) erratic movement (side view). The paths were generated by the EthoVision XT tracking software.

considered when interpreting zebrafish motor-related activity. From our observations, when compared to the control zebrafish, which exhibited normal continuous swimming, the MPTP-injected zebrafish exhibited abnormal swimming phenotypes, such as looping (distinct circular motion), freezing (complete movement cessation), and erratic movement (abrupt changes in velocity or direction as well as recurrent rapid darting) (Fig 8).

Despite the known regenerative ability of the zebrafish and the transient effect of MPTP neurotoxin on the zebrafish, we did not observe any tendency for the altered swimming behaviour to recover on the 96th hour after injection. Perhaps, a longer observation period is required for the recovery to show a discernible effect.

## Changes in body weights after the administration of MPTP

Apart from the swimming behaviour, we also analysed the differences in the initial and final body weights of the experimental zebrafish. It is well known that experimental procedures, such as injection and prolonged isolation, may induce stress on zebrafish, leading to weight loss due to decreased appetite. Nevertheless, no significant difference was observed between the initial and final body weights of both MPTP-injected and control zebrafish (Fig 9).

## Discussion

Emphasizing the induction of neurotoxin onto adult zebrafish to establish a reliable animal PD model, this study evaluated the effect of MPTP administration on the swimming behaviour of adult zebrafish. One dose of MPTP was delivered through i/p injection to adult zebrafish and its effects on swimming behaviour were examined in an open field test at three time points (0h, 24h, and 96h-post injection). The results of our study revealed that MPTP administration

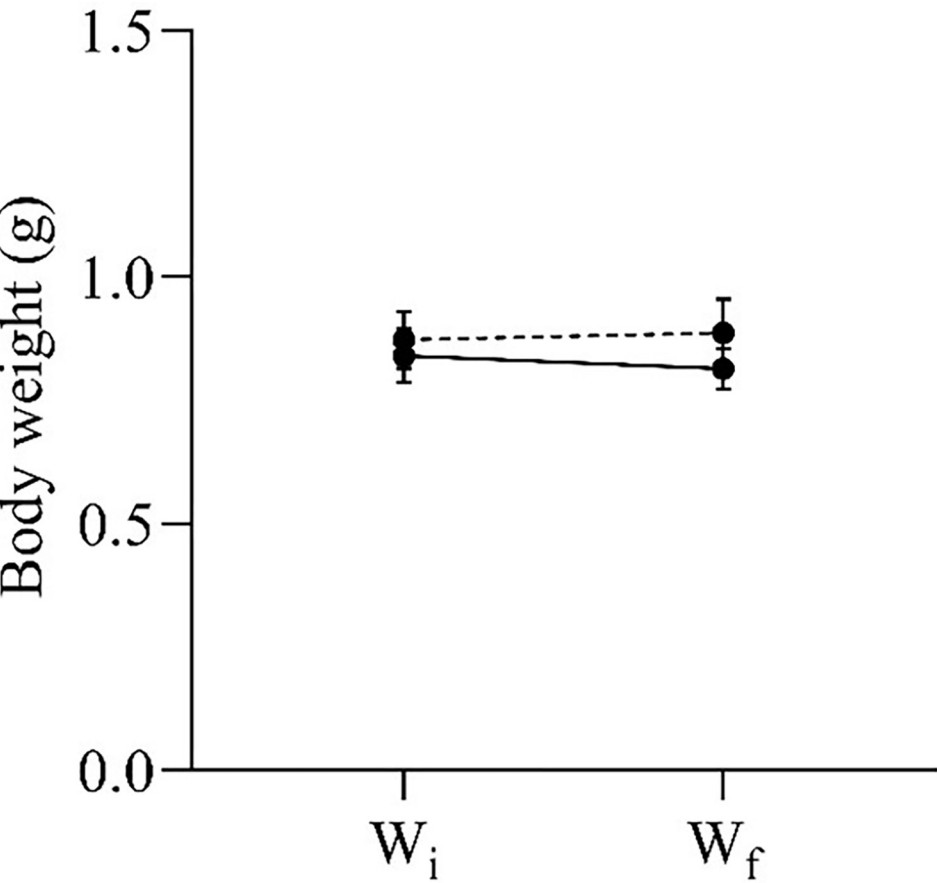

**Fig 9. Initial ($W_i$) and final ($W_f$) body weights of MPTP-injected (solid lines) and control (dashed lines) zebrafish.**
Two-way RM ANOVA revealed that there was no statistically significant interaction between the effects of MPTP
injection and time on body weight changes (F (1,18) = 1.24, p = 0.281). All values are presented as mean and error bars
represent the SEM. n = 10 per group.

into adult zebrafish reduced the locomotor activity, altered the complexity of the swimming
pattern, and induced abnormal swimming phenotypes. Besides that, the body weight changes
in response to MPTP administration was investigated. Throughout the experimental period,
we found that the body weights of both MPTP-injected and control zebrafish were unchanged.

Zebrafish is a well-known model for PD. The zebrafish ventral diencephalon (vDn) is
thought to reflect the human substantia nigra since it is dominated by dopaminergic neurons.
Furthermore, Du et al. [27] detected a dopaminergic projection from the vDn to the subpal-
lium region in zebrafish larvae, suggesting that this projection is equivalent to the nigrostriatal
dopaminergic pathway of humans. In terms of locomotion, zebrafish exhibit sophisticated
behaviours reflective of PD. A number of studies have reported the effects of MPTP on zebra-
fish locomotion [18, 19, 21, 25, 30], but only a few have performed prolonged observation [21,
25]. Assessing the behavioural changes at several time points is critical as it provides stronger
evidence for the establishment of MPTP-induced zebrafish model of PD.

In this study, marked alterations in locomotor activity, indicated by slower swimming
velocity, shorter distance travelled, and longer freezing maintenance, was detected 24 hours
after the MPTP administration and still remained discernible 96 hours later. This implies that,
compared to the control zebrafish, the MPTP-injected zebrafish swam slower and had periods
of idleness. It is also fair to note that, the number of freezing bouts decreased from the 24th to

the 96th hour following injection. This indicates that, on the 96[th] hour, instead of alternating between swimming and freezing, the MPTP-injected zebrafish were predominantly immobile during the whole duration of the behavior test. Our finding is in line with those of previous studies, which documented hypolocomotion in neurotoxin-induced animal models [25, 28–30]. Anichtchik et al. [25] found a substantial decrease in locomotor activity and an increase in turn angle in MPTP-injected zebrafish on the 1[st], 3[rd], 6[th], and 9[th] days following injection. However, the study lacks information on other swimming phenotypes such as meander and angular velocity. Furthermore, MPTP-injected zebrafish spent more time in the IN zone, according to the study. In contrast, we found that MPTP-injected and control zebrafish spent the same amount of time in the IN zone. The study did not, however, measure the number of crossings between the IN and OUT zones, which is more representative of the swimming pattern than time spent in the IN zone.

Aside from the locomotor activity, how the zebrafish swim is also a robust indicator of neurotoxic effects [31]. In this study, we observed that the swimming pattern of MPTP-injected zebrafish was different from that of control zebrafish. In terms of the time spent at the center of the tank, both groups showed similar duration, however, the control zebrafish crossed the center more frequently than the MPTP-injected zebrafish. An open field tank offers a new area for exploration [32]. In this context, a higher frequency of crossing the inner zone indicates zebrafish exploratory behaviour [33]. We interpreted that this reflects a regular swimming pattern. In contrast, the MPTP-injected zebrafish preferred to swim nearby their initial position and avoided frequent crossing of the center of the tank and remained in the same zone throughout the behaviour test period. Therefore, the tendency of the MPTP-injected zebrafish to stay in only one zone during the behavior test period suggests that the neurotoxin may have an effect on their exploratory behavior.

Another parameter that reflects the motor aspects of the zebrafish is the turning rate, which comprises of the turn angle, meander, and angular velocity. The turn angle is an important aspect in assessing fish behaviour as it is a measure of maneuverability achieved by muscular contraction of the tail [34]. Mechanically, the body must perform more work to navigate a larger turn angle [35]. Aside from being largely studied in the context of predatory or aversive stimuli, turn angle also corresponds with erratic movements [36]. In our study, MPTP-injected zebrafish had a slightly larger turn angle 24 hours after injection and a two-fold increase on the 96[th] hour compared to controls. Meander on the other hand, measures the directionality of the fish [37]. Similar to the turn angle, a larger meander value is associated with erratic movements [32]. An increase in meandering was reported in adult zebrafish exposed to acute acrylamide [38] and mercury [39] neurotoxicity. Comparably, our MPTP-injected zebrafish had more than three times the meander values of controls at 24h and 96h post-injection, suggesting impaired directionality caused by MPTP neurotoxicity. Finally, when comparing MPTP-injected zebrafish to controls, we found that their angular velocity, or turning speed, was significantly higher. Most notably, after 96 hours following injection, the treated zebrafish had double the turning speed of controlled fish. Angular velocity, like meander, is used to assess the directionality of the fish [40]. Reportedly, a high angular velocity correlates to significant bending of zebrafish entire body, which reflects erratic movements [41, 42]. Collectively, the turn angle, meander, and angular velocity of the zebrafish after MPTP administration were profoundly increased on the 24[th] hour and kept increasing on the 96[th] hour post-injection, resulting in fast and sharp changes in swimming direction. This behaviour could be analogous to the incoordination seen in human PD patients when executing voluntary movements. The cardinal motor features of PD include bradykinesia (slowness of movement) and impaired voluntary motor coordination [43] support the assertion that our zebrafish model of PD mimics human PD.

Most studies documented the path taken and the swimming speed, but they neglected to address the swimming phenotypes linked to neurotoxin-induced neuropathy. In our study, several different types of swimming were documented that were exclusively seen in the MPTP-injected group, but not in the control group, such as freezing, looping, and erratic movement. Freezing is characterized as a complete cessation of movement, except for gills and eyes [31]. MPTP-injected zebrafish in our study froze for the majority of the time throughout the behaviour test period. Notably, these zebrafish showed no signs of increased respiration, suggesting that the immobility was caused by the neurotoxic agent rather than stress or anxiety. Looping, on the other hand, is defined as a distinct circular motion [31]. Zebrafish is considered to be swimming in a loop when it is spinning around a virtual point outside the body, in contrast to circling which is characterized as an orbiting behaviour around a specific object or subject [31]. Repeated looping behaviour was observed in our MPTP-injected zebrafish, with a noticeable increase in velocity during the spinning motion. Looping behaviour in fish indicates uncontrolled movement due to motor impairment [31, 44]. Looping is also associated with defective optic nerve projection which leads to visual system impairment [45]. Evidently, MPTP administration was found to cause retinal degeneration, a vital component that communicates with the optic nerve [17]. Apoptotic photoreceptors and inner retina neuronal loss were observed in mice treated with MPTP [17]. The optokinetic response (OKR) assay, which uses alternating dark and light concept, is particularly useful for assessing zebrafish visual-motor response and visual acuity [46]. Perhaps, this could account for the looping behaviour seen in our MPTP-injected zebrafish. However, in this study, we did not perform the OKR assay. Another abnormal swimming phenotype evident in the MPTP-injected zebrafish was erratic movement, which is defined as recurrent rapid darting with sharp directional changes [31]. This pattern of swimming indicates motor incoordination, which is usually associated with neurotoxicity [31, 47].

In humans, the dopaminergic neuron in the SNpc modulates movement control through direct and indirect basal ganglia pathways [4]. Unfortunately, in PD, due to the significant dopaminergic neuronal loss, dopamine input from the SNpc is substantially reduced, and thus, disrupts the pathways. A transgenic mice model harboring the PD genotype showed diminished firing of nigral dopaminergic neurons and consequent loss of striatal dopamine signaling, resulting in movement impairment [48]. Likewise, MPTP has also been linked to the death of dopaminergic neurons. It has the ability to mimic the specific degeneration of nigral dopaminergic neurons seen in the development of PD. According to Sarath Babu et al. [18] and Selvaraj et al. [19], reduced locomotor activity in MPTP-induced animal models is comparable to human bradykinesia and dyskinesia, suggesting a disruption in motor control ability.

In terms of MPTP pharmacokinetics in zebrafish, intraperitoneally injected MPTP enters the zebrafish peritoneal cavity. Due to its small size ($MW_{MPTP}$: 173.25 g/mol), MPTP is subsequently absorbed into the portal vein via mesenteric capillaries by diffusion. From the portal vein, it is transported to the brain through the circulatory system [49] and easily crosses the blood-brain barrier (BBB) [50]. Inside the brain, it is converted to MPP+ by monoamine oxidase B (MAO B), which is found in glial cells. The MPP+ is transported into dopaminergic neurons via dopamine transporter (DAT) protein due to its similar structure with dopamine [51–53]. The presence of MPP+ inside the dopaminergic neurons is toxic, as it impairs mitochondrial electron transport chain, leading to mitochondrial dysfunction [54], oxidative stress [55], neuroinflammation [56], and ultimately neuronal death. Conclusively, MPTP administration impairs movement control by degenerating dopaminergic neurons needed to modulate the movement pathways.

Kalyn & Ekker [21] reported a substantial reduction of dopaminergic neurons in the telencephalon and olfactory bulbs of adult zebrafish following MPTP treatment. Similarly, a rodent

study by Alam et al. [57] documented a drop in nigral tyrosine hydroxylase (TH)-positive signals after MPTP induction, which indicates a reduction in dopaminergic neuronal density in the substantia nigra of MPTP-injected mice. Moreover, the same study detected a decrease in striatal dopamine concentrations, indicating a disruption in dopaminergic signaling between the substantia nigra and striatum regions of the treated mice [57]. Therefore, the alterations in swimming behaviours of the zebrafish observed in our study could be due to the dysregulation of the movement signaling pathway caused by MPTP-induced loss of dopaminergic neurons.

When assessing the effect of neurotoxin on animals, it is also useful to measure the difference between the final and initial body weights of the animals. It is well known that body weight loss is correlated with stress level [58]. Hence, information on the body weights may portray how stressful were the animals with the experimental regime. In our study, we found no significant differences in body weights before and after the experiment, indicating that both MPTP-injected and control zebrafish had normal appetites throughout the experimental period. This suggests that the i/p injection and the 96 hours isolation regime did not trigger stress in the zebrafish. Even so, further investigation should be carried out to support this claim, such as measuring the levels of stress hormone and the expression of stress-related genes.

While our data are preliminary, they imply that the time taken for MPTP neurotoxin to induce observable PD-like symptoms in adult zebrafish is around 24 hours after the intraperitoneal injection and the effect still persisted 96[th] hour post-injection. This information is critical because most PD research utilizing MPTP-induced zebrafish model often aim to produce treatments or therapies that can potentially reduce or reverse PD symptoms. Hence, it is important to ensure that the MPTP effect is already stabilized, and persistent, otherwise confounding factors could skew the results and cause misinterpretation. Despite the known regenerative ability of the zebrafish and the transient effect of MPTP neurotoxin on the zebrafish, we did not observe any tendency for the altered swimming behaviour to recover on the 96[th]h hour after injection. Perhaps, a longer observation period is required for the recovery to show a discernible effect. Also, a recent study delivered MPTP neurotoxin into adult zebrafish through intracerebroventricular (icv) injection and reported hypolocomotion effect [21]. It is fair to note that, our findings corroborate the observation seen in their study, demonstrating that the i/p injection technique is comparable to the more advanced icv injection technique.

Our study focused solely on the behavioural aspect of MPTP on adult zebrafish. Future works will include correlating the behavioural profile with molecular aspects, such as genomic and proteomic networking, as well as histochemical changes in the brain of MPTP-injected zebrafish, to obtain more sophisticated information on how MPTP actions on the brain translate into behavioural presentations. Also, we observed the behaviour only at three time points (0h, 24h, and 96h post-injection). To observe more gradual changes and long-term MPTP effect, smaller time intervals (for example, 12-hourly) and longer observation of the behaviour test should be conducted. Plus, further work can be done to compare the swimming behaviour between male and female zebrafish to study sex differences on MPTP-induced locomotor impairment. Notwithstanding these limitations, our study offers some insights into the swimming behaviour, in terms of locomotor activity and swimming pattern, of the MPTP-induced zebrafish model of PD.

## Conclusions

Taken together, our findings demonstrate that a single systemic injection of MPTP (200 μg/g bw) into adult zebrafish causes significant reduction in locomotor activity, changes in swimming pattern, and abnormal swimming phenotypes, but no significant body weight changes.

One of the more significant conclusions to emerge from this study is that MPTP effects on adult zebrafish can be distinctively seen after 24 hours of its administration and still persisted 96 hours later. The second key conclusion is that, in addition to decreased locomotor activity, MPTP administration provoked motor incoordination in adult zebrafish, as seen by the complexity of swimming pattern and abnormal swimming phenotypes in MPTP-injected zebrafish. Further investigations are needed to determine the MPTP mechanism of action that causes these changes.

## Acknowledgments

We gratefully acknowledge the IIUM Central Research and Animal Facility (CREAM) for assisting with zebrafish husbandry and supplying data analysis equipment, particularly the Noldus EthoVision XT tracking software.

## Author Contributions

**Conceptualization:** Khairiah Razali, Wael M. Y. Mohamed.

**Data curation:** Khairiah Razali, Wael M. Y. Mohamed.

**Formal analysis:** Khairiah Razali, Mohd Hamzah Mohd Nasir, Wael M. Y. Mohamed.

**Funding acquisition:** Wael M. Y. Mohamed.

**Investigation:** Mohd Hamzah Mohd Nasir.

**Methodology:** Khairiah Razali.

**Project administration:** Khairiah Razali.

**Resources:** Wael M. Y. Mohamed.

**Software:** Mohd Hamzah Mohd Nasir.

**Supervision:** Noratikah Othman, Abd Almonem Doolaanea, Jaya Kumar, Wael M. Y. Mohamed.

**Validation:** Wisam Nabeel Ibrahim, Wael M. Y. Mohamed.

**Writing – original draft:** Khairiah Razali.

**Writing – review & editing:** Khairiah Razali, Mohd Hamzah Mohd Nasir, Noratikah Othman, Abd Almonem Doolaanea, Jaya Kumar, Wisam Nabeel Ibrahim, Wael M. Y. Mohamed.

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
