## [Decision Letter · Decision Letter 0]

25 Jul 2022

PONE-D-22-16443A Stable Five Days of Hypolocomotion and Abnormal Swimming Pattern in 1-Methyl-4-Phenyl-1,2,3,6-Tetrahydropyridine (MPTP)-Induced Zebrafish Model of Parkinson’s DiseasePLOS ONE

Dear Dr. Mohamed,

Thank you for submitting your manuscript to PLOS ONE. After careful consideration, we feel that it has merit but does not fully meet PLOS ONE’s publication criteria as it currently stands. Therefore, we invite you to submit a revised version of the manuscript that addresses the points raised during the review process.

Dear Dr Wael Mohamed,

You will find enclose 4 reviews for your manuscript. They all find merit to your work. From their commentaries, you will see that no additional experiments are presently required, but we will ask you to carefully take into consideration all their questions and remarks, and more precisely to rigorously document responses to reviewers 3 and 4 comments.

Sincerely,

Prof Bruno Giros, PhD

McGill University

Academic Editor

We look forward to receiving your revised manuscript.

Kind regards,

Bruno Giros, Ph.D.

Academic Editor

PLOS ONE

Journal Requirements:

Reviewers' comments:

Reviewer's Responses to Questions

**Comments to the Author**

1. Is the manuscript technically sound, and do the data support the conclusions?

Reviewer #1: Partly

Reviewer #2: Yes

Reviewer #3: Yes

Reviewer #4: Yes

2. Has the statistical analysis been performed appropriately and rigorously? 

Reviewer #1: Yes

Reviewer #2: Yes

Reviewer #3: Yes

Reviewer #4: Yes

3. Have the authors made all data underlying the findings in their manuscript fully available?

Reviewer #1: Yes

Reviewer #2: Yes

Reviewer #3: Yes

Reviewer #4: No

4. Is the manuscript presented in an intelligible fashion and written in standard English?

Reviewer #1: Yes

Reviewer #2: Yes

Reviewer #3: Yes

Reviewer #4: Yes

5. Review Comments to the Author

Reviewer #1: The work presented by Khairiah Razali et al. aims at describing the impact of MPTP treatment in zebrafish. They analyzed the effect of the drug on different locomotor assays in zebrafish before and after 24h and 96 h. They suggested that the treatment induced locomotor deficits that may be relevant to Parkinson’s disease. This topic is important since there is a lack of in vivo preclinical model that are mimicking human pathology, thus pertaining the development of therapeutic strategies. Overall, the experiment are well conducted and the results are of importance.

However, I have some recommendations.

1) The authors claim that treatment with MPTP would induce visual deficit that could explain some of the locomotor behavior. In order to address this important question, I would suggest the authors to realize optokinetic response experiment to determine if MPTP treatment affects visual acuity.

2) The observed phenotype with the turn angle, the meander and the angular velocity necessitate prosper discussion with relevant citations in order to strengthen their conclusion on the Parkinson’s disease phenotype.

Reviewer #2: A good manuscript.

Title needs some rephrasing

Discussion needs to address better some other toxicological and biochemical mechanism involved in this

Results are decent and well worked

Introduction looks ok.

Methodology also fine, with no flaws detected.

Reviewer #3: General comments (page numbers refer to the numbers at the bottom of the docx-file, line numbers to the numbers on the left margin of the document):

In the submitted MS Razali and coworkers report the behavioral change by a single dose of MPTP intraperitoneally administered to the adult zebrafish 1 and 4 days after injection. This is a methodologically sound and thorough study and the novel insight is basically the stability of behavioral changes between these time points, except a loss of increased freezing bouts at 4 days (see also specific point to page 12 below). This finding might be important for the use of the MPTP-model in zebrafish in the context of neuroprotective studies considering the well-known reversibility of the neurodegeneration induced by MPTP in this species. Due to the focus of this rather methodologically circumscribed study, all discussions of mechanistic aspects of MPTP in the zebrafish are redundant (page 18, line 417 to page 20, line 458), which means that the discussion should be shortened considerably and shifted to a more thorough comparison of the specific behavioral findings to that reported in the literature (e.g. comparison to the study of Anichtchik OV et al., J. Neurochem. 2004).

Specific points:

Page 3, line 54:

Who says that cholinergic neurons are lost in the substantia nigra in PD?

Page 3, line 55:

A review on the promise Zebrafish in PD is not an appropriate review on the basic characteristics of the pathology of PD

Page 8, line 176-182 or page 9, line 196-202:

Please specify the geometry of the IN and OUT zone within the recording tank.

Page 9, line 196-202:

Please specify the definition of bouts in the context of freezing behavior.

Page 9, line 196-202:

Please specify the definition of frequency in the IN zone in the context of swimming pattern.

Page 12, line 262-264, page 12, line 273-275 (legend to Fig.5):

Is there no statistical significance between 96h and 24h in Fig.5b for MPTP-injected fish?

Page 13, line 287 (legend to Fig.6):

Please specify for frequency in the IN zone during which time the numbers are counted (similarly to the way the number of freezing bouts are described in the legend to Fig.5b).

Page 18, line 422/423:

The wording suggests that reference 36 (Miskitsh & Chacko, 2014) is reporting data about MPTP, which is not the case

Page 18, line 424 to page 19, line 429:

It is unclear why just reference 37 (Risiglione et al., 2022) is especially relevant for the degenerating effect of MPTP on dopaminergic neurons, a neurotoxic mechanism elucidated by studies decades earlier.

Page 19, line 440-441:

Reference 38 (Zinger et al., 2011), a review about neuroinflammation and kynurenine in PD, is not an appropriate reference for the indirect pathway in basal ganglia, first described decades earlier.

Page 19, line 450-452, page 20, line 454-456:

Reference 41 (Huang et al. Parkinson’s disease 2017) is not a specific rodent study, but a review about changes in the MPTP mouse model.

Page 23-26, References:

Please give more complete information for cited books.

Reviewer #4: This study investigates the neurobehavioral toxicity of a single i.p. injection of MPTP, a model chemical for Parkinson's disease, in adult zebrafish. The main merit of this study is a complete characterization of behavioral endpoints at 0, 24h, and 96h post MPTP injection. The behavioral analysis was rather comprehensive and in great details. The main pitfalls are 1) both male and female adult fish were used in this study, but no mention about whether there was any sex difference in all these behavioral parameters; and 2) lack of histopathological examination of the MPTP-treated brain to confirm any DA neuron loss, though authors acknowledged this in their Discussion. A minor comment is that there are a few recent MPTP studies using the zebrafish were not cited and incorporated in the Discussion, for example: https://pubs.acs.org/doi/abs/10.1021/acschemneuro.2c00089

https://www.sciencedirect.com/science/article/abs/pii/S0161813X22000997?via%3Dihub

6. PLOS authors have the option to publish the peer review history of their article (what does this mean?). If published, this will include your full peer review and any attached files.

Reviewer #1: No

Reviewer #2: No

Reviewer #3: No

Reviewer #4: No

---

## [Author Response · Author response to Decision Letter 0]

15 Aug 2022

We appreciate the time spent by reviewers to read our MS. We did our best to address all their comments for improving of the MS. We uploaded a clean revised version of the MS

---

## [Decision Letter · Decision Letter 1]

5 Sep 2022

Characterization of Neurobehavioral Pattern in A Zebrafish 1-Methyl-4-Phenyl-1,2,3,6-Tetrahydropyridine (MPTP)-Induced Model: A 96-hour Behavioral Study

PONE-D-22-16443R1

Dear Dr. Mohamed,

We’re pleased to inform you that your manuscript has been judged scientifically suitable for publication and will be formally accepted for publication once it meets all outstanding technical requirements.

Kind regards,

Bruno Giros, Ph.D.

Academic Editor

PLOS ONE

Additional Editor Comments (optional):

Reviewers' comments:

Reviewer's Responses to Questions

**Comments to the Author**

1. If the authors have adequately addressed your comments raised in a previous round of review and you feel that this manuscript is now acceptable for publication, you may indicate that here to bypass the “Comments to the Author” section, enter your conflict of interest statement in the “Confidential to Editor” section, and submit your "Accept" recommendation.

Reviewer #1: All comments have been addressed

Reviewer #2: All comments have been addressed

Reviewer #3: All comments have been addressed

Reviewer #4: All comments have been addressed

2. Is the manuscript technically sound, and do the data support the conclusions?

Reviewer #1: Yes

Reviewer #2: Yes

Reviewer #3: Yes

Reviewer #4: Yes

3. Has the statistical analysis been performed appropriately and rigorously? 

Reviewer #1: Yes

Reviewer #2: Yes

Reviewer #3: Yes

Reviewer #4: Yes

4. Have the authors made all data underlying the findings in their manuscript fully available?

Reviewer #1: Yes

Reviewer #2: Yes

Reviewer #3: Yes

Reviewer #4: Yes

5. Is the manuscript presented in an intelligible fashion and written in standard English?

Reviewer #1: Yes

Reviewer #2: Yes

Reviewer #3: Yes

Reviewer #4: Yes

6. Review Comments to the Author

Reviewer #1: (No Response)

Reviewer #2: Accept. The answers are ok. Ms looks ok for publication now, in my humble opinion.

Accept. The answers are ok. Ms looks ok for publication now, in my humble opinion.

Reviewer #3: All my suggestions were met by appropriate changes. The Discussion increased by about 30 lines; if the Editor considers that a problem, the mechanistic aspects that I already considered redundant in my review of the original submission could be shortened, e.g. page 21, line 474-line 485.

Reviewer #4: The authors have adequately addressed all my previous comments. I have no further comments for this revised version.

7. PLOS authors have the option to publish the peer review history of their article (what does this mean?). If published, this will include your full peer review and any attached files.

Reviewer #1: No

Reviewer #2: No

Reviewer #3: No

Reviewer #4: No

---

## [Editor Report · Acceptance letter]

22 Sep 2022

PONE-D-22-16443R1 

Characterization of Neurobehavioral Pattern in A Zebrafish 1-Methyl-4-Phenyl-1,2,3,6-Tetrahydropyridine (MPTP)-Induced Model: A 96-hour Behavioral Study 

Dear Dr. Mohamed:

I'm pleased to inform you that your manuscript has been deemed suitable for publication in PLOS ONE. Congratulations! Your manuscript is now with our production department. 

Kind regards, 

on behalf of

Professor Bruno Giros 

Academic Editor

PLOS ONE